# Prioritisation of Adverse Drug Events Leading to Hospital Admission and Occurring during Hospitalisation: A RAND Survey

**DOI:** 10.3390/jcm11154254

**Published:** 2022-07-22

**Authors:** Annette Haerdtlein, Anna Maria Boehmer, Katharina Karsten Dafonte, Marietta Rottenkolber, Ulrich Jaehde, Tobias Dreischulte

**Affiliations:** 1Institute of General Practice and Family Medicine, University Hospital, LMU Munich, 80336 Munich, Germany; annette.haerdtlein@med.uni-muenchen.de (A.H.); marietta.rottenkolber@med.uni-muenchen.de (M.R.); 2Doctoral Program Clinical Pharmacy, University Hospital, LMU Munich, 81377 Munich, Germany; 3Department of Clinical Pharmacy, Institute of Pharmacy, University of Bonn, 53121 Bonn, Germany; annaboehmer@uni-bonn.de (A.M.B.); u.jaehde@uni-bonn.de (U.J.); 4Institute of Clinical Chemistry and Clinical Pharmacology, University of Bonn, 53127 Bonn, Germany; katharina.karsten_dafonte@ukbonn.de

**Keywords:** adverse drug events, drug-related side effects, consensus, RAND survey, prioritisation, medication safety

## Abstract

(1) Adverse drug events (ADEs) are a common cause of emergency department visits and occur frequently during hospitalisation. Instruments that facilitate the detection of the most relevant ADEs could lead to a more targeted and efficient use of limited resources in research and practice. (2) We conducted two consensus processes based on the RAND/UCLA appropriateness method, in order to prioritise ADEs leading to hospital admission (panel 1) and occurring during hospital stay (panel 2) for inclusion in future ADE measurement instruments. In each panel, the experts were asked to assess the “overall importance” of each ADE on a four-point Likert scale (1 = not important to 4 = very important). ADEs with a median rating of ≥3 without disagreement were defined as “prioritised“. (3) The 13 experts in panel 1 prioritised 38 out of 65 ADEs, while the 12 experts in panel 2 prioritised 34 out of 63 ADEs. The highest rated events were acute kidney injury and hypoglycaemia (both panels), as well as Stevens–Johnson syndrome in panel 1 and rhabdomyolysis in panel 2. (4) The survey led to a set of ADEs for which there was consensus that they were of particular importance as presentations of acute medication-related harm, thereby providing a focus for further medication safety research and clinical practice.

## 1. Introduction

Adverse drug events (ADEs) are a common cause of emergency department visits and also occur frequently during hospitalisation [1,2]. Recent prospective observational studies on acute emergency department or hospital admissions have shown that about five to 30% were attributable to ADEs, of which two thirds or more were assumed to be at least possibly preventable [3,4,5,6]. According to a systematic review and meta-analysis of in-hospital ADEs, 19% of inpatients suffer from an ADE and approximately one third of these ADEs are judged as preventable [2]. Such ADEs can impose a significant burden on patients [1,7,8,9,10]. For example, in a prospective study of patients that were admitted to four large hospital emergency departments due to suspected adverse drug reactions (ADRs), 4% of patients with ADRs died and 1% suffered permanent damage [1]. A systematic review and meta-analysis of the characteristics of ADRs revealed that 31% of ADRs occurring in hospitalised older adults are severe [7]. Another meta-analysis reported an overall percentage of drug-related deaths among inpatients of around 6% [10]. ADEs are also associated with major economic challenges for health care systems [11,12]. In Germany, the estimated total costs for ADE-related emergency hospitalisations may amount to EUR 2.25B per year [12]. Inpatient ADEs were estimated to increase the average treatment costs per patient by EUR 970 [11].

Preventing harm from medication requires identification of the risks before harm occurs; existing risk detection tools range from software highlighting drug–drug interactions to lists of potentially inappropriate medication (PIM) [13,14,15,16,17]. Nevertheless, even in settings implementing the most sophisticated countermeasures, the continued occurrence of ADEs is inevitable. In this scenario, the detection of ADEs and their causes is crucial to ameliorate harm and to prevent recurrence [18]. ADE recognition is also a pre-requisite to avoiding unintended prescribing cascades (i.e., the prescription of new drugs to treat ADEs that are misinterpreted as new medical conditions) [19].

The gold standard of ADE detection is a causality assessment by clinical experts using validated algorithms, but this is time consuming and requires experience [20,21]. The development of screening instruments, which can be efficiently applied at the point of care to identify potential ADEs, would therefore be an important step forward. If implemented in routinely collected electronic data sources, such screening instruments could also be used in clinical surveillance or research to repeatedly measure changes in the occurrence of potential ADEs at scale [17,22].

The aim of the present study was to identify a set of prioritised ADEs as a basis for defining medication safety measures for applications in clinical practice (e.g., decision support), clinical surveillance and research (e.g., as outcome measures).

The research that is reported here is embedded in the Germany-wide POLAR (POLypharmacy, drug interActions, Risks) project, which is part of the Medical Informatics Initiative (MII). POLAR focusses on the use of routinely collected hospital inpatient data to detect and prevent medication-related problems including ADEs at hospital admission and during hospitalisation [23].

## 2. Materials and Methods

### 2.1. Study Design

We conducted two separate expert consensus processes based on the RAND/UCLA appropriateness method (RAM) [24], in order to prioritise adverse drug events (ADEs) for two clinical settings: (1) at hospital admission (to prioritise ADEs originating in ambulatory care); (2) during hospital stay (to prioritise ADEs originating in the hospital). Similar to the Delphi method, we developed an assessment form with candidate ADEs for each panel based on a literature search. Panellists then independently rated each ADE on two occasions, with the first-round ratings fed back to them before the second-round ratings were placed. In contrast to the Delphi method but consistent with the RAM, a moderated face-to-face meeting (one for each panel) was held in between rounds to enable the exchange of arguments between experts (Figure 1).

### 2.2. Selection of Experts

For both expert panels, we recruited physicians and pharmacists with an academic interest or clinical experience in the detection or treatment of ADEs at hospital admission or during hospital stay, respectively, aiming for a balanced distribution of the two professions and of self-reported (predominant) professional activity as scientists or clinicians in each panel. Aiming for approximately 12 experts in each panel, we initially invited a total of 36 physicians and pharmacists that were involved in the POLAR project, as well as 14 additional physicians and pharmacists who had either been involved in large German studies on inpatient or outpatient ADEs or were nominated by POLAR project experts. Those that were interested in participating completed a self-declaration form about their field of activity, and their professional and academic background.

### 2.3. Design of the Assessment Forms

In order to generate a comprehensive list of candidate ADEs occurring at hospital admission or during inpatient stay, respectively, two systematic literature searches were performed. We searched MEDLINE for articles that were published between 01/2000 and 07/2020, combining search terms for the setting (‘hospital’ or ‘hospital admission’ in ‘Germany’) and the focus of the study (‘ADEs’). All empirical studies reporting ADEs in the general population on admission or during inpatient stay were included, whereas studies targeting specific populations or ADEs were excluded. More details of the literature search are provided in Appendix A.

From selected publications, we extracted all reported ADEs, classified them by organ system, and grouped them into superordinate categories, guided by the International Classification of Diseases, 10th Revision (ICD-10). Since the focus was on the detection of acute events at hospital admission or during hospital stay, ADEs that were unlikely to lead to hospital admission or worsen acutely (e.g., osteoporosis or obesity) were excluded. In addition, we excluded the following ADEs since they were out of scope: explicit consequences of surgical or medical procedures (e.g., infections after infusion, transfusion or injection) or drug poisoning (e.g., harmful use of non-addictive substances); events only involving children (e.g., neonatal icterus or laryngospasm) or pregnant women (e.g., liver diseases during pregnancy, childbirth and the postpartum period); and ADEs that were judged to be the consequences of medication underuse (e.g., uncontrolled pain). All exclusions were based on consensus discussions within the core research team (AH, TD, AMB, UJ). The studies that were used to generate the lists of ADEs to be rated by each panel were summarised in two different evidence reports, i.e., one for each setting-specific panel.

### 2.4. Pretest and Optimisation of the Assessment Forms

Based on the literature searches, drafts of the assessment forms (which were virtually identical for both panels) and the two different evidence reports were pretested and optimised in two stages. In the first stage, a convenience sample of three pharmacists and one physician (who were not part of the research team) were presented with the draft assessment form and the evidence document for panel 2 (hospital stay). In stage 2, another two pharmacists and two physicians were presented with a revised assessment form and the evidence document for panel 1 (hospital admission). Feedback from the pretest participants was obtained at each stage via semi-structured interviews (interview guide: Appendix A) which focussed on the comprehensiveness of the ADEs that were listed in the assessment forms, the comprehensibility of the scales, rating instructions and ADE descriptions; on the grouping of ADEs in superordinate categories; and the comprehensiveness and utility of the evidence reports. Expert feedback emerging from the first stage was implemented and a last round of amendments after the second feedback round yielded the final optimised versions of the first-round assessment forms.

### 2.5. Rating Process

#### 2.5.1. Definitions and Pre-Specifications

“Overall importance” was the pre-specified key criterion that was used to prioritise the ADEs. Since we considered “overall importance” an insufficiently specific concept to be uniformly understood by the panellists, we asked them to rate for each ADE the importance to “conduct a medication review (in the near future) as a strategy to prevent further or repeated harm” in relation to an average patient (Figure 2). In addition, we asked the panellists to rate each ADE for “seriousness” (defined as the likelihood of the ADE leading to serious harm (prolonged hospital stay, permanent damage or life-threatening condition)) and “drug-relatedness” (defined as the likelihood that one or more drug(s) contributed to the adverse event). Although prioritisation was to be solely based on overall importance ratings, the additional rating scales served the dual purpose of (a) encouraging the panellists to consistently consider these aspects in their overall importance ratings; (2) identifying sources of disagreement between the panellists to inform discussions prior to the second-round ratings. Given that the same ADEs may be worded in ways that reflect different levels of severity (e.g., constipation and ileus), the panellists were instructed that all the ADEs to be rated would be assumed to be sufficiently severe to warrant hospital admission (panel 1) or medical treatment (panel 2). For laboratory parameters (e.g., hyperkalaemia), threshold values were provided to specify severity. For broader ADEs or those that were identified as potentially ambiguous during pretests, examples were provided for clarity. We also pre-specified that ADEs with a median overall importance rating of ≥3 without disagreement would be defined as “prioritised”. Disagreement was pre-defined to be present if at least 30% of expert ratings were 1 or 2 (for items with a median of ≥3 consistent with prioritisation), or 3 or 4 (for items with a median of <2 consistent with non-prioritisation).

#### 2.5.2. Rating Rounds

The experts were sent the assessment form by e-mail, including rating instructions and the respective evidence report for each setting. Approximately two weeks after completion of the first round, a face-to-face expert meeting took place for each panel, moderated by TD for panel 1 (hospital admission) and UJ for panel 2 (hospital stay), respectively. At the beginning and during discussions, important aspects to consider were highlighted, including clarification that all ratings should be placed in relation to the ADE being caused by a drug, rather than by underuse of drugs or drug withdrawal.

For each ADE, the first-round ratings were summarised (median and distributions of ratings for overall importance, seriousness and drug-relatedness, and whether there was disagreement) and fed back to the experts. To facilitate the discussion, ADEs were discussed in thematic blocks (e.g., cardiovascular ADEs, gastrointestinal ADEs). The focus of discussion was on ADEs with disagreement regarding their overall importance after the first rating round, while differences in seriousness and drug-relatedness were used to inform the discussion.

After discussion of a thematically related set of ADEs, the panellists directly placed their second-round ratings. The ADEs with a median overall importance rating of ≥3 without disagreement (defined as above) after the second-round rating were deemed “prioritised”.

## 3. Results

### 3.1. Expert Panels

The expert panels comprised 13 members from 11 German university sites (panel 1) and 12 members from nine German university sites (panel 2), respectively. Table 1 shows that members of both panels were approximately balanced in terms of professional background and main field of professional activity. The majority of the recruited experts had additional research or clinical qualifications.

### 3.2. Literature Search and Design of Round 1 Assessment Forms

For panel 1 (hospital admission), the first-round assessment form was informed by nine publications and for panel 2 (hospital stay) by eight publications (a flow chart of identified, screened, included and excluded publications is provided in Appendix A). The extracted ADEs of both literature searches led to the same 74 superordinate events within 13 organ classes to be included in the first drafts of the assessment forms for both panels, of which 57 ADEs satisfied our inclusion and exclusion criteria (excluded ADEs are listed in Appendix A). The main changes that emerged from the experts’ feedback were to include more detailed rating instructions and to split the ADEs that were considered too broad for assessment. For example, the ADE bone marrow suppression was split into anaemia, thrombocytopenia and neutropenia/agranulocytosis. Additionally, in order to optimally adapt the assessment forms to the respective setting, the ADEs myopathy (without rhabdomyolysis) and somnolence were only assessed in panel 1 (hospital admission). The resulting round 1 assessment forms contained 63 ADEs (panel 1) and 61 ADEs (panel 2), respectively (Figure 3).

### 3.3. Rating Process and Findings

For both panels, the ratings for round 1 are provided in Appendix A and the results of round 2 are presented in Figure 4 and Figure 5.

#### 3.3.1. Panel 1: Prioritisation of ADEs on Admission

The round 1 assessment form was emailed to the panellists in January 2021 and the expert panel met, discussed the first-round findings and conducted the second-round ratings on 26 February 2021.

All 13 (100%) experts returned a fully completed round 1 assessment form, all took part in the moderated expert discussion and returned a fully completed round 2 assessment form. In round 1, of 63 ADEs assessed, there was consensus for 31 (49%) to be prioritised (median ≥ 3 without disagreement) and for seven (11%) not to be prioritised (median < 3 without disagreement). Disagreement was present for 25 ADEs (40%) (disagreement on prioritisation: 11 ADEs; disagreement on non-prioritisation: 14 ADEs, Figure 3). During the discussion, the experts decided to split the ADE gastroenteritis and colitis (into pseudomembranous colitis and gastroenteritis, and colitis excluding pseudomembranous colitis) and the ADE urinary retention (to be assessed for people aged <65 and ≥65 years separately). Therefore, a total of 65 ADEs were rated in round 2. Of these, there was consensus to prioritise 38 ADEs (58%) and not to prioritise 18 (28%). The second assessment round resolved first round disagreements for 16 ADEs (five of which were now prioritised and 11 not prioritised). However, after the second rating round, disagreement remained for 9 ADEs (14%) (four on prioritisation, five on non-prioritisation).

#### 3.3.2. Panel 2: Prioritisation of ADEs during Inpatient Stay

The round 1 assessment form was emailed to the panellists in January 2021 and the expert panel met on 4 March 2021 to discuss the first-round findings and conduct the second-round ratings.

All 12 (100%) experts returned the round 1 assessment form, all took part in the moderated expert discussion and 11 experts (92%) returned a fully completed round 2 assessment form. Of 61 ADEs rated in round 1, there was consensus for 25 ADEs (41%) to be prioritised (median ≥ 3 without disagreement) and for 2 ADEs (3%) not to be prioritised (median < 3 without disagreement). Disagreement was present for 34 ADEs (56%) (disagreement on prioritisation: 19 ADEs; disagreement on non-prioritisation: 15 ADEs, Figure 3). As in panel 1 (hospital admission), the panel 2 (hospital stay) experts decided to split the ADE gastroenteritis and colitis as above and to additionally split the ADE thyroid dysfunction into the ADEs hyperthyroidism and hypothyroidism. After round 2, there was consensus to prioritise 34/63 ADEs (54%) and not to prioritise 13 (21%). The second rating round resolved disagreements for 19 ADEs (8 were now prioritised and 11 not prioritised). However, after the second rating round, disagreement remained for 16 ADEs (25%) (eight on prioritisation; eight on non-prioritisation).

### 3.4. Comparison of the Overall Importance Findings in Panels 1 and 2

A comparison of the panel findings (Figure 3 and Figure 4) shows that 29 ADEs were prioritised in both panels, while nine ADEs were prioritised only in panel 1 (hospital admission), and four ADEs were prioritised only in panel 2 (hospital stay). In both panels, acute kidney injury and hypoglycaemia were among the three highest rated events, which also featured Stevens–Johnson syndrome in panel 1 (hospital admission) and rhabdomyolysis in panel 2 (hospital stay).

### 3.5. Relationship between Overall Importance, Seriousness and Drug-Relatedness

Appendix A show an overview of the assessment results for overall importance, seriousness and drug-relatedness after the second-round ratings of both panels. Of ADEs with median overall importance ratings of ≥3, 22/42 ADEs (52%) in panel 1 (hospital admission) and 26/42 ADEs (62%) in panel 2 (hospital stay) also had median ratings of ≥3 for both seriousness and drug-relatedness. Nevertheless, there were eight ADEs (12%) in panel 1 (hospital admission) and one ADE (2%) in panel 2 (hospital stay) where ratings for seriousness and drug-relatedness diverged from the overall importance rating (overall importance rating ≥ 3 and the other ratings < 3), namely other allergic skin reactions, hallucinations, hypotension, uncontrolled hyperglycaemia, urinary retention (≥65 years), myopathy, thyroid dysfunction and acute gout attack in panel 1 (hospital admission) and hyperthyroidism in panel 2 (hospital stay). A total of nine ADEs (14%) in panel 1 (hospital admission) and 13 ADEs (21%) in panel 2 (hospital stay) had a median overall importance rating of <3, but a median seriousness rating of ≥3 (e.g., acute coronary syndrome and cerebral infarction in both panels), while there were no ADEs with a median overall importance rating of <3 and a median drug-relatedness rating of ≥3 in both panels.

## 4. Discussion

### 4.1. Summary of Findings

In this study, we identified a total of 38/65 (58%) ADEs at hospital admission and 34/63 (54%) ADEs during hospital stay, for which there was consensus on their high overall importance, thus classified as “prioritised”. While the majority of prioritised ADEs after round 2 were common to both panels (n = 29), a total of 13 ADEs were selected only by one panel (nine ADEs only by panel 1 and four ADEs only by panel 2), which supports our approach of separate setting-specific consensus processes.

The median importance rating was ≥2 for all the ADEs in both panels, which may reflect that all the ADEs that were included in the assessment form had previously been empirically identified as potential presentations of medication-related harm and emphasises the relevance of the ADEs that were included in the assessment form. Despite this, our study indicates that by asking the panellists to rate the importance of conducting a medication review to prevent further or repeated harm from the ADE on a 4-point Likert scale, it is possible to discriminate between more and less relevant ADEs.

The ratings for overall importance on the one hand, and for seriousness and drug-relatedness on the other, generally pointed in the same directions. Of the ADEs with median overall importance ratings of ≥3, the majority (≈60%) also had median ratings of ≥3 for seriousness and drug-relatedness in both panels, which suggests that seriousness and drug-relatedness are important drivers for overall importance. However, the finding that among ADEs with lower overall importance ratings (<3), all had a lower drug-relatedness rating (<3)—whereas several had a higher seriousness rating (≥3)—suggests that drug-relatedness may be a more important driver of overall importance than seriousness.

There were several examples where the ratings for either seriousness or drug-relatedness diverged from the overall importance ratings, suggesting that other criteria may also play a role. For example, despite an overall importance rating of 3, the ADE myopathy had a median score of <3 for both seriousness and drug-relatedness. Myopathy is multi-causal (which may explain a lower drug-relatedness rating) and rhabdomyolysis was rated separately (so that lower seriousness ratings may be explained by myopathy being limited to less severe presentations). Nevertheless, myopathy is a common adverse reaction of frequently prescribed drugs (i.e., statins) and early recognition may prevent more serious events [25]. This suggests that the prevalence of ADEs and the preventability of further drug-related harm may be independent drivers of overall importance.

The different settings caused diverging prioritisation in the respective panels for some ADEs, partly due to differences between the drugs that are used in outpatient and inpatient settings. For example, the ADE toxic damage to the inner ear is predominantly caused by aminoglycosides, which are almost exclusively used in the inpatient setting [26]. This likely explains why this ADE was prioritised by panel 2 (hospital stay), but not by panel 1 (hospital admission).

### 4.2. Comparison with Literature

To the best of our knowledge, this is the first consensus process study to prioritise ADEs systematically at both hospital admission and inpatient stay. There is only one similar expert survey from the United States by Jeon et al., focussing on inpatient ADEs, which exclusively prioritised ADEs that were deemed as preventable by pharmacist intervention [27]. We included ADEs irrespective of their preventability because our focus was on their detection to avert further harm, or to measure them in the context of clinical surveillance or research. The survey by Jeon et al. identified 21 ADEs as priorities for preventive action by pharmacists. Of the latter, the following comparable ADEs of our assessment form were not prioritised by panel 2 (which also focussed on ADEs originating in hospital): thrombosis, nausea and vomiting, hypothyroidism, hypertensive crisis, decompensated heart failure, anaemia and gastrointestinal ulcers (although gastrointestinal bleeding was prioritised in our set). These differences are likely explained by our exclusion of ADEs that are the consequence of the underuse of medication.

### 4.3. Strengths and Limitations

A strength of the present RAND consensus process is the heterogeneous composition of the two expert panels, with a balanced representation of physicians and pharmacists who are predominantly involved in scientific research or clinical practice and have expertise in studying or detecting ADEs at hospital admission or during inpatient stay. Also noteworthy is their distribution across numerous university sites throughout Germany, so that the expert panels covered a breadth of experience from a variety of perspectives. We systematically tested and optimised the assessment forms (two iterations) prior to their distribution. This meant that any ambiguities of rating constructs or wording of ADEs could be minimised. Any remaining misunderstandings were clarified during moderated discussions. The personal discussions during the panel meetings also enabled an exchange of arguments and experiences for the panellists to consider in their second-round ratings, which are key strengths of the RAND consensus process. Due to the simultaneous implementation of the consensus process for two different settings, a direct comparison of the results was possible, revealing commonalities and differences in the relevance of ADEs at hospital admission and during inpatient stay.

A limitation of our RAND consensus process was that for feasibility reasons, the large number of individual ADEs had to be combined into superordinate categories, partly resulting in broad ADE definitions. Nevertheless, we compensated for this by providing definitions and examples, thereby ensuring that all experts had the same basis for assessment. In addition, where ADEs appeared to be too heterogeneous to be rated collectively, the panellists had the opportunity (and made use of it) to suggest splitting ADEs during expert discussions, which were then rated separately in the second rating round.

### 4.4. Implications for Research and Practice

The two sets of prioritised ADEs that are developed here can provide a basis for a number of future applications.

In order to support clinical practice, the prioritised ADEs could be implemented in routine electronic data sources as decision support and/or case finding tools to prompt medication reviews and/or to efficiently direct staff resources, e.g., of clinical pharmacists or pharmacologists. The aim here would be to prevent further or repeated harm from detected ADEs (i.e., ADE management and secondary prevention). Our ADE lists therefore supplement the work by Jeon et al., who prioritised ADEs that could be prevented by clinical pharmacists (i.e., primary prevention of ADEs) [27].

In order to support clinical research, the prioritised ADEs could be implemented in routine electronic data sources to efficiently and repeatedly measure the prevalence or incidence of ADEs, both to inform and evaluate quality improvement interventions. Instruments to efficiently and specifically measure the clinical impact of medication safety initiatives are currently missing. While there are examples of interventional studies, which have measured drug-related hospital admissions (ascertained by expert assessment), most of them have been either limited to measuring processes (i.e., medication use) or unspecific outcomes, such as all-cause hospital admissions or the prolongation of inpatient stay [28,29,30]. Our prioritised lists of ADEs may therefore provide a basis to fill an important gap in the medication safety literature.

For applications in both clinical practice and research, the sensitivity and specificity of ADE detection instruments are important considerations. A lack of sensitivity could result in missing ADE cases that require management, while a lack of specificity may lead to alert fatigue in a clinical context and a limited responsiveness to change in a research context. Although we have identified adverse drug events that may be important presentations of medication-related harm, many of the events can also have other causes, which implies limitations in specificity.

In order to increase the specificity of ADE detection, previous authors have combined adverse events with preceding suboptimal medication use patterns (e.g., hospital admission for gastrointestinal bleeding preceded by the use of antiplatelets in patients aged ≥75 years without gastroprotection) [31,32]. This approach focusses on failures in the medication use process but limits the sensitivity of ADE detection because it misses all unpreventable ADEs and cannot identify all preventable ADEs (since the spectrum of suboptimal medication use patterns is too broad to be comprehensively pre-specified). A potentially more promising compromise between sensitivity and specificity is to combine the ADEs that are identified here with potentially causative drugs (e.g., hospital admission for gastrointestinal bleeding preceded by the use of antiplatelets), which would restrict detected adverse drug events to those where a drug-related cause is (at least) possible without a restriction to pre-specified medication use patterns.

## 5. Conclusions

By conducting a RAND survey for the two clinical settings ‘hospital admission’ and ‘hospital stay’, we have identified two sets of ADEs for which there is consensus that they are of particular importance as presentations of acute medication-related harm, thereby providing a focus for further medication safety research and clinical practice. As part of the POLAR project, we aim to further develop the prioritised items into indicators of potential ADRs by identifying potentially causative medication in a second consensus process. The indicators will be implemented in data that are routinely available in the data integration centres of German University hospitals.

## Figures and Tables

**Figure 1 jcm-11-04254-f001:**
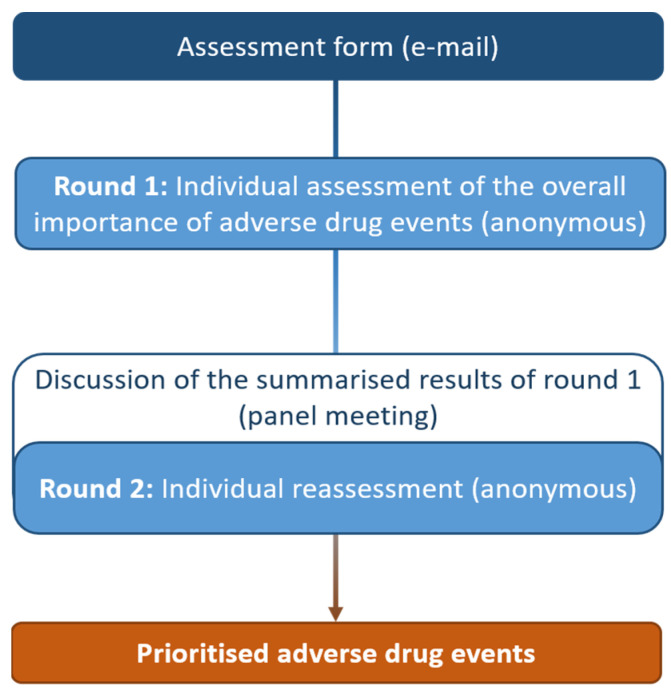
RAND consensus process followed by each of the two expert panels prioritising adverse drug events (ADEs) on admission (panel 1) and during inpatient stay (panel 2).

**Figure 2 jcm-11-04254-f002:**
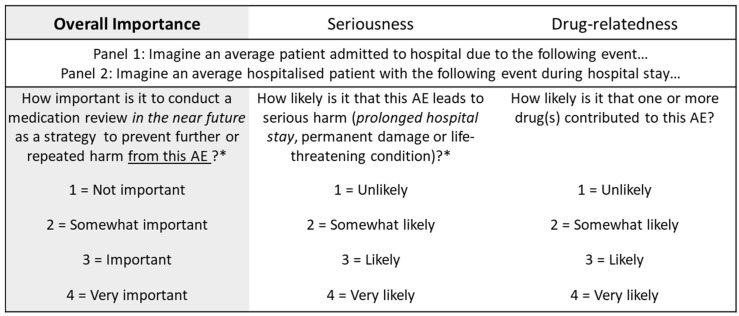
Assessment criteria and rating scales for panel 1 (hospital admission) and panel 2 (hospital stay); * *Italic* terms were only part of the assessment form of panel 2. *Abbreviations:* AE = adverse event.

**Figure 3 jcm-11-04254-f003:**
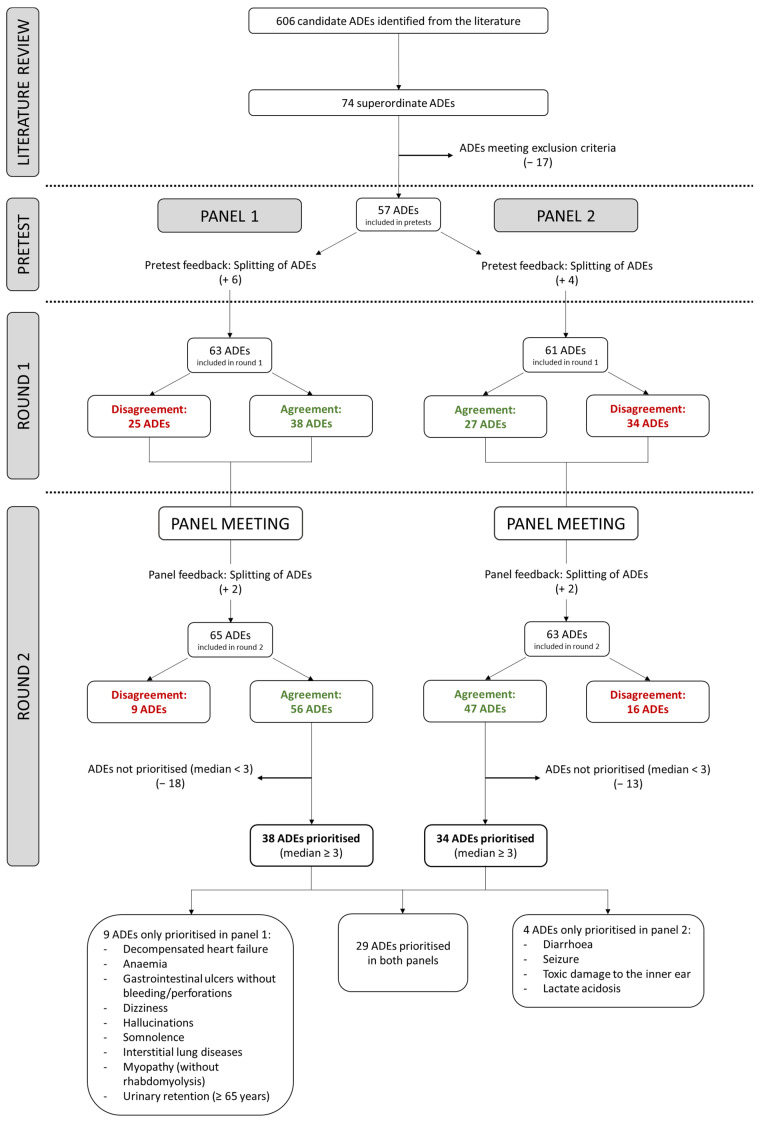
Flow chart showing the ADE prioritisation process in panel 1 (hospital admission) and panel 2 (hospital stay). *Abbreviations:* ADEs = adverse drug events.

**Figure 4 jcm-11-04254-f004:**
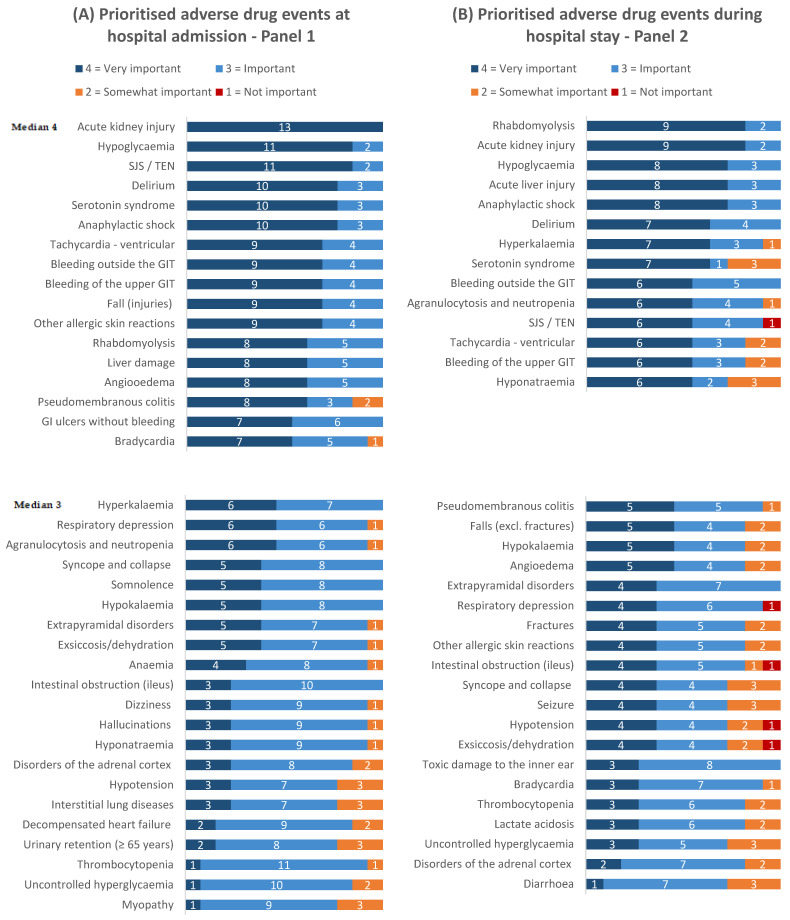
Distributions of overall importance ratings in (**A**) panel 1 (hospital admission) and (**B**) panel 2 (hospital stay) for the prioritised adverse drug events after completion of the second assessment round of the RAND consensus process. *Abbreviations:* SJS/TEN = Stevens–Johnson syndrome/toxic epidermal necrolysis; GIT = gastrointestinal tract.

**Figure 5 jcm-11-04254-f005:**
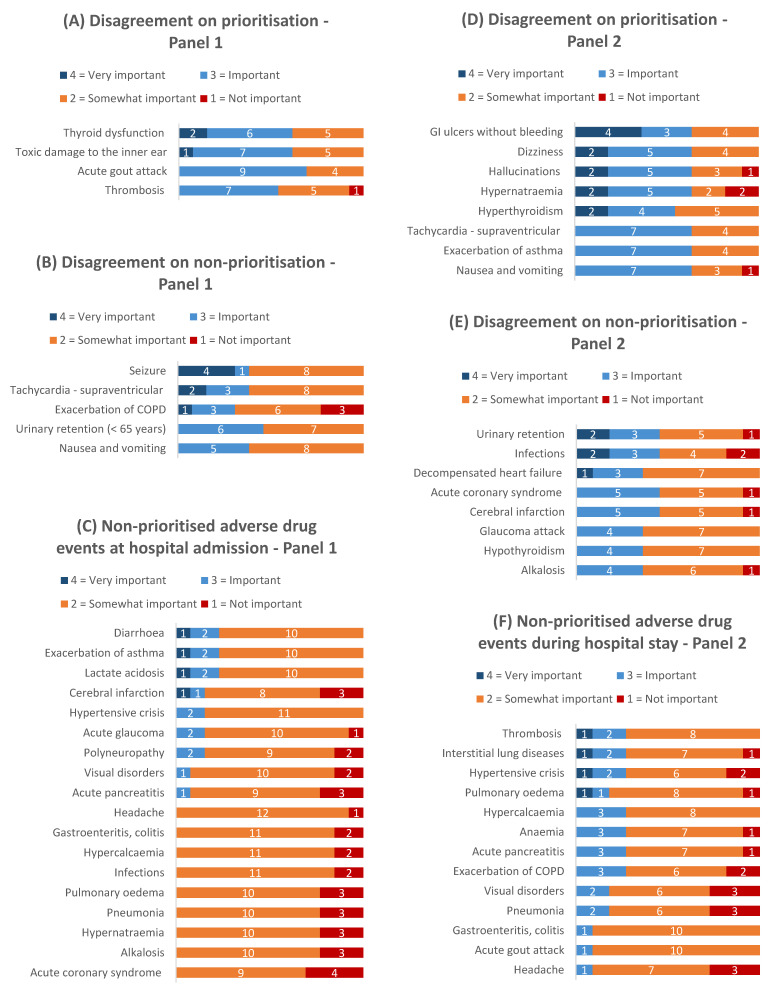
Distributions of overall importance ratings in (**A**–**C**) panel 1 and (**D**–**F**) panel 2 for ADEs that were not prioritised after the second assessment round, incl. those that reached no consensus. *Abbreviations:* COPD = chronic obstructive pulmonary disease; GI = gastrointestinal.

**Table 1 jcm-11-04254-t001:** Characteristics of participating experts in the two panels.

	Panel 1 (Hospital Admission) n = 13		Panel 2 (Hospital Stay) n = 12	
	Physicians (n = 6)	Pharmacists (n = 7)	Physicians (n = 6)	Pharmacists (n = 6)
**Academic background**				
Additional qualification (habilitation/doctorate and/or clinical specialist qualification)	6 (100%)	4 (57%)	4 (67%)	5 (83%)
**Main field of professional activity**				
Scientific research	4 (67%)	2 (29%)	3 (50%)	1 (17%)
Clinical practice	0 (0%)	2 (29%)	1 (17%)	2 (33%)
Both	2 (33%)	3 (43%)	2 (33%)	3 (50%)

## Data Availability

The data presented in this study are available in Figure 4 and Figure 5, Appendix A.

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
