# Peer review of "Prioritisation of Adverse Drug Events Leading to Hospital Admission and Occurring during Hospitalisation: A RAND Survey"

_jcm, 2022, doi:10.3390/jcm11154254_

Round 1

Reviewer 1 Report

Congratulations on your paper. I have some considerations that may help to improve the study.

Abstract: Correct the word "hypoglycemia". 

Introduction: I strongly recommended update the references on the introduction. There is a lot of important papers about ADE in the last five years that could be cited.

Methods: The study lacks information about when the rounds of panels were conducted. It is important for a complete understanding of how the process of analysis occurred to provide details about the period of each step of the study.

Discussion: I suggest the change of "4.1. Summary of findings" for the results section. The discussion section needs to be complemented with other studies that helps the reader to fully understand the applicability of ADE prioritization conducted. If “the focus here is on their detection to avert further harm or to measure them in the context of clinical surveillance or research” studies that support the affirmation for the hospital context must be provided. Therefore: how the ADE prioritized can be used for defining medication safety measures for applications in clinical practice (e.g. decision support), clinical surveillance and research (e.g. as outcome measures)?

Author Response

Dear Reviewer,

thank you very much for your valuable feedback. We have carefully revised our manuscript in response to your comments and recommendations and hope we have addressed them to your satisfaction. Please find attached the revised manuscript and responses. We are looking forward to hearing from you.

Kind regards,

Tobias Dreischulte

Reviewer 2 Report

Dr. haerdtlein and dr Boehmer and collaborators, you reported a concensus of ADEs of particular importance at hopital admission and during inpatient stay.

The manuscript is very clear and well-written, the topic of adverse event is of importance. The methods are well described and appropriate, the results are supported by the data presented and the discussion is fair-balanced while citing up-to-date references.

Overall, I have on this occasion minor comments that could improve the scientific value and would like to congratulate the authors for their hard job on this important issue.

- in abstract: add the number of experts in each panel : the13 experts of panel 1 prioritised 38 out of 74? ADEs and the 12 experts of panel 2 ...

- in introduction, l46 : EUR 970 per hospitalization? per year?

- in Materials: p3 l106 : did you think to use MedDRA dictionnary? or other?

l 116: abbreviations? and later p 4 l 167

p 4 l 142: add prolonged hospitalization stay

- Results : i propose to modify figure 3, for an easier comprehension and maybe discussion about the interest of the second round to reduce disagreements (enclosed).

- "seriousness" and "relatedness" appears as secondary outcomes, results are not objectively presented (seems like a discussion) (for exemple "generally point"). and discussion is like a repetition.

Discussion: there are some repetitions of the results.

Author Response

Dear Reviewer,

thank you very much for your valuable feedback. We have carefully revised our manuscript in response to your comments and recommendations and hope we have addressed them to your satisfaction. Please, find attached the revised manuscript and responses. We are looking forward to hearing from you.

Kind regards,

Tobias Dreischulte
